# Osteoporotic hip fracture—Comorbidities and factors associated with in-hospital mortality in the elderly: A nine-year cohort study in Brazil

**Viviane Cristina Uliana Peterle** [1,2]*, **Maria Rita Carvalho Garbi Novaes**[1,2], **Paulo Emiliano Bezerra Junior**[1], **João Carlos Geber Júnior** [3], **Rodrigo Tinôco Magalhães Cavalcante**[1], **Jurandi Barrozo da Silva Junior**[1], **Ray Costa Portela**[1,4], **Ana Patricia de Paula**[1,2]

1 Escola Superior de Ciências da Saúde (Escs/Fepecs), Brasília, DF, Brazil, 2 Universidade de Brasilia, Brasília, DF, Brazil, 3 Department of Internal Medicine, Hospital das Clinicas HCFMUSP, Faculdade de Medicina, Universidade de São Paulo, São Paulo, Brazil, 4 Department of Surgery, Mayo Clinic, Rochester, Minnesota, United States of America

* vivianepeterle@hotmail.com

**Data Availability Statement:** All relevant data are within the paper and its Supporting Information files.

## Abstract

### Introduction

The aim of the study was to identify factors associated with the causes of in-hospital morbidity and mortality in an elderly Brazilian population due to osteoporotic hip fractures.

### Method

Retrospective cohort study involving a population over 60 years of age admitted to hospital due to osteoporotic hip fractures and followed up from hospitalization to outcome (discharge or mortality) from 2010 to 2018, in a public hospital in Brasília, the capital of Brazil. Multivariate analysis was performed using the Poisson regression model with a robust variance, observing the hierarchical model proposed and the receiver operating characteristic (ROC) curve to obtain the cutoff point for mortality incidence in relation the total length of hospital stay. Significance level was set as p < 0.05. The analyses were conducted using the SAS 9.4 software.

### Result

The mean hospital mortality rate among the 402 patients involved was 18.4%, and the associations made with the outcome mortality were per relevance: respiratory infection, age over 90 years, high preoperative cardiovascular risk, chronic obstructive pulmonary disease (COPD) as comorbidity, serum hemoglobin level ≤ 10 and other infections. Mortality also showed association with longer total length of hospital stay, as well as with prolonged postoperative period.

### Conclusion

Hip fractures in the elderly due to osteoporosis indicate a relationship between the sicker profile of the aging elderly population and the prevalence of chronic diseases strongly

**Funding:** MRCGN Grant number: 0006400000044/
2019-21 Funder: Fundação de Ensino e Pesquisa
em Ciências da Saúde http://www.fepecs.edu.br/
Yes. Concecpetualization, Funding acquisition,
Metodology, Project administration, Supervision.

**Competing interests:** The authors have declared
that no competing interests exist.

associated with in-hospital infections, contributing to increased mortality. There were fewer
early interventions, and mortality was also associated with prolonged postoperative period.
The aim of this study was not to compare independent variables with each other, but suggests the relationship between the presence of comorbidities, which predisposes to the
development of infections, directly linked to mortality.

## Introduction

Hip fractures due to a fall from standing height are related to bone fragility and can be used to
diagnose osteoporosis [1]. The factors associated with falls and risk of fractures and with the
process of osteoporosis onset indicate a relationship between longevity and chronic diseases
[2].

A study using data from the Brazilian National Health Survey (PNS) conducted in 2013 in
Brazil revealed that the prevalence of three simultaneous diseases in the age group of 60 years
or more was 3.7-fold higher than in the aged group 35 to 59 years and almost 20-fold higher
than in those between 18 and 34 years old [3].

Recent studies have shown that isolated comorbidities are a dominant predictor of mortality [4, 5], being strongly associated with in-hospital infections [6]. In turn, infection is associated with a substantially increased mortality risk [7].

The surgical decision-making process for hip fracture repair in the elderly is not straightforward. The surgeon can use several tools to determine the mortality risk and define who would
be more benefitted from surgical interventions and who should be referred for non-surgical
interventions [8].

Mortality prediction models, which describe the outcomes distribution among the population with a particular set of characteristics, can support physicians in adapting treatments for
decision making in frail elderly patients, as can causal effect estimates, which help us understand the impact of different treatment decisions made in that population [9, 10].

The aim of this study was to identify factors associated with in-hospital mortality among
patients with osteoporotic hip fractures undergoing surgery or conservative treatment considering an elderly population with multiple comorbidities treated in Brazil, a developing country
undergoing a rapid population aging process. This is the first Brazilian study to consider the
risk attributable to the identified factors.

## Methods

### Data sources and configuration

Retrospective cohort study involving patients with 60 years of age or more admitted to hospital
due to hip fracture after a fall from standing height and followed up from hospitalization to
outcome (discharge or mortality) from 2010 to 2018.

The study was conducted in a reference public hospital in the treatment of orthopedic
trauma and accredited by the Brazilian Society of Orthopedics and Traumatology (SBOT)
located in Brasília, the capital of Brazil. The hospital has protocols in place for prophylaxis
against deep-vein thrombosis [11].

Data were collected from medical records of patients during the hospital stay evolution and
complementary examinations. Exclusion criteria were pathological fractures or undefined
minimal trauma fractures.

For identification and description of the variables, the data were collected through the patients' admission records in the Orthopedics ward in the mentioned period (2010–2018) through Track-care@ electronic medical records. After that, each clinical record related to the identified patient was analyzed individually and sequentially in the daily record of the nursing team for data conference, where the selection of the variables of interest to the study and transcription to a database on Microsoft Excel for further statistical analysis (S1 Database and S1 File).

The mortality outcome was considered a dependent variable. The independent variables collected were (a) demographic: sex, age (60 to 79, 80 to 89, > 90 years); (b) factors associated with clinical conditions and comorbidities: hemoglobin ($\leq$ 10 g/dL, > 10 g/dL), systemic arterial hypertension (SAH) (no, yes), diabetes mellitus (DM) (no, yes), neurological disorders (no, yes), chronic obstructive pulmonary disease (COPD) (no, yes), surgical risk (low to moderate, high), femur fracture (intracapsular, extracapsular); and (c) in-hospital variables: type of surgery (did not undergo surgery, osteosynthesis, arthroplasty), respiratory infection (no, yes), urinary tract infection (no, yes), another type of infection (no, yes) and (none, one, two or more), pulmonary thromboembolism (PTE) (no, yes), intensive care unit (ICU) stay in days ($\leq$ 3, > 3).

An anatomical subdivision into intracapsular fracture (fracture of head and neck of the femur–S72.0) and extracapsular fracture (pertrochanteric fracture–S72.1 and subtrochanteric fracture–S72.2) was used to classify the hip fracture areas according to the International Classification of Diseases, Tenth Revision (ICD-10).

In regard to the identification of comorbidities, the selection was based on the etiological or topographic diagnosis in order to better identify the nature of the condition and its association with the mortality outcome instead of a classification score. SAH and DM were independently identified by standardized classifications [12, 13]. Concerning the infections identified during hospital stay, sites of involvement such as the lungs, urinary tract, and others were determined.

The use of the Detsky Modified Cardiac Risk Index (1986) was standardized by stratifying the patients into three risk groups (Class 1: 0–15 points (low risk); Class 2: 20–30 points (moderate risk); and Class 3: > 30 points (high risk)) [14].

Due to the varied clinical conditions, the Wells score, which considers the main risk factors without using complementary examinations, was used to estimate the clinical probability of PTE. The final sum provides an approximate patient classification as high (> 7 points), medium (2–6 points), or low (0–1 point) [15–18].

To prevent biases such as information bias, the researchers trained the technicians responsible for collecting data and controlling database entries; also, a periodical analysis and ongoing review for data inconsistency were carried out throughout the study period.

The study was approved by the Research Ethics Committee under number CAAE: 89658718.8.0000.5553, and the informed consent form was waived upon approval.

## Statistical analysis

The statistical analysis was initiated with the descriptive analysis of the variables' frequencies and determination of the incidences associated with mortality, with the respective confidence intervals (CI).

To test the effect of the independent variables on mortality, a multivariate analysis was conducted using the Poisson regression model with robust variance, observing the following hierarchical model: demographic variables composed the first stage of analysis, preoperative variables composed the second stage, and in-hospital variables composed the third stage. In this analysis, variables with a p < 0.10 were considered adjustment factors for subsequent blocks within each hierarchical level.

The Poisson regression model with robust variance was chosen because it provides a better estimate of incidence ratios, which represent the measures of effect for prospective studies, such as the relative risk (RR), in a more significant manner. The analysis took place in two phases, bivariate and hierarchical multiple regression analysis, and their RRs and respective 95% CI were calculated [19, 20].

Tolerance indicator values of less than 0.403 were the limit of multicollinearity set among independent variables.

For the quantitative time variables, the Shapiro-Wilk test was used to test the data normality, while for bivariate analysis, the non-parametric Mann-Whitney test was used to compare the groups (mortality and survival outcomes).

The receiver operating characteristic (ROC) curve was calculated to obtain the prevailing cutoff point for mortality in relation to the total length of hospital stay and to the postoperative period. The cutoff point was obtained by combining two criteria: the first, defined as the shortest Euclidean distance between the binary classification result estimated by the test and the point that provides the perfect predictor (100% sensitivity and 100% specificity); and the second, defined as the maximum Euclidean distance between the binary classification result estimated by the test and the point that provides a non-informative predictor (45° straight line).

P-values of $p < 0.05$ were considered significant. The analyses were conducted using the SAS 9.4 software.

## Results

Four hundred and two patients who met the selection criteria were included, and the mean mortality rate as an in-hospital outcome for the period was 18.4%. The total number of elderly patients admitted to hospital with hip fractures due to falls, the distribution of the mean in-hospital mortality rate, and the mean length of hospital stay in days, per year, are described in Table 1.

The descriptive analysis is described in Table 2, showing the characteristics of patients sustaining fractures and mortality according to the studied variables. As for the demographic variables, a predominance of fractures in women (n = 260; 64.68%), but higher mortality in men (20.42%; 95% CI 13.76–27.08), was found. The age group 80–89 years had a higher incidence of fractures (n = 142; 32.34%); however, the highest proportional mortality was in the age group over 90 years (52%; 95% CI 30.67–57.79).

Regarding the preoperative assessment variables, most patients had a low to moderate cardiac risk (n = 268; 66%), although a high number of patients (n = 353; 87.81%) presented with some comorbidity at the time of hospital admission. SAH (n = 305; 75.87%) and DM (n = 139; 34.58%) were the most frequent comorbidities. Almost half of the patients (n = 218; 54.23%) also had other comorbidities concomitant with those selected for analysis, such as chronic kidney disease (6%), hypothyroidism (8%), and dyspeptic disorders (5%). As for the type of fractures, the intracapsular (femoral neck) ones presented a slight predominance (n = 218; 54.23%).

**Table 1. Total patients with hip fracture admitted, mean in-hospital mortality rate and mean length of hospital per year.**

| Admission year | 2010 | 2011 | 2012 | 2013 | 2014 | 2015 | 2016 | 2017 | 2018 | Total |
|---|---|---|---|---|---|---|---|---|---|---|
| Patients admitted (n) | 45 | 60 | 40 | 60 | 51 | 54 | 31 | 36 | 25 | 402 |
| In-hospital mortality | 9 | 8 | 8 | 11 | 13 | 12 | 5 | 4 | 4 | 74 |
| Mean in-hospital mortality rate (n, %) | 20.0% | 13.3% | 20.0% | 18.3% | 25.5% | 22.2% | 16.1% | 11.1% | 16.0% | 18.4% |
| Mean length of hospital stay (days) | 23.4 | 20.1 | 28.2 | 21.6 | 27.2 | 30.0 | 27.5 | 28.4 | 25.5 | 25.4 |

**Table 2. Characteristic of patients and mortality.**

| Variables | | Patients (n = 402) | Porcentage (%) | Mortality (%) | 95% IC |
|---|---|---|---|---|---|
| **Sex** | Women | 260 | 64.68 | 17.31 | 12.69–21.93 |
| | Men | 142 | 35.32 | 20.42 | 13.76–27.08 |
| **Age** | 60–69 | 78 | 19.40 | 5.13 | 0.21–10.04 |
| | 70–79 | 130 | 32.34 | 8.46 | 3.66–13.27 |
| | 80–89 | 142 | 35.32 | 25.35 | 18.17–32.54 |
| | > 90 | 52 | 12.94 | 44.23 | 30.67–57.79 |
| **Hemoglobin** | ≤ 10 | 87 | 21.64 | 28.74 | 19.19–38.28 |
| | > 10 | 315 | 78.36 | 15.56 | 11.54–19.57 |
| **Comorbidities** | No | 49 | 12.19 | 8.16 | 0.46–15.86 |
| | Yes | 353 | 87.81 | 19.83 | 15.65–24.01 |
| **SAH** | No | 97 | 24.13 | 12.37 | 5.79–18.95 |
| | Yes | 305 | 75.87 | 20.33 | 15.79–24.86 |
| **DM** | No | 263 | 65.42 | 16.35 | 11.86–20.85 |
| | Yes | 139 | 34.58 | 22.30 | 15.35–29.25 |
| **Neurological disorders** | No | 296 | 73.63 | 15.88 | 11.70–20.06 |
| | Yes | 106 | 26.37 | 25.47 | 17.14–33.80 |
| **COPD** | No | 371 | 92.29 | 15.90 | 12.17–19.64 |
| | Yes | 31 | 7.71 | 48.39 | 30.72–66.05 |
| **Surgical risk** | Low to moderate | 268 | 66.67 | 8.58 | 5.21–11.95 |
| | High | 134 | 33.33 | 38.06 | 29.80–46.32 |
| **Femur fracture** | Intracapsular | 218 | 54.23 | 18.81 | 13.60–24.02 |
| | Extracapsular | 184 | 45.77 | 17.93 | 12.37–23.50 |
| **Type of surgery** | Non-surgical | 92 | 22.89 | 38.04 | 28.08–48.01 |
| | Osteosynthesis | 173 | 43.03 | 13.87 | 8.70–19.05 |
| | Arthroplasty | 137 | 34.08 | 10.95 | 5.70–16.20 |
| **Infection** | No | 267 | 66.42 | 2.25 | 0.46–4.03 |
| | Yes | 135 | 33.58 | 50.37 | 41.90–58.84 |
| **Respiratory infection** | No | 316 | 66.42 | 4.43 | 2.15–6.71 |
| | Yes | 86 | 21,39 | 69.77 | 60.02–79.51 |
| **Urinary tract infection** | No | 338 | 84.08 | 14.20 | 10.46–17.94 |
| | Yes | 64 | 15.92 | 40.62 | 28.54–52.71 |
| **Other infection** | No | 386 | 96.02 | 17.36 | 13.56–21.15 |
| | Yes | 16 | 3.98 | 43.75 | 19.34–68.16 |
| **PTE** | No | 384 | 95.52 | 16.41 | 12.69–20.13 |
| | Yes | 18 | 4.48 | 61.11 | 38.48–83.73 |
| **Days in ICU** | ≤ 3 | 322 | 80.10 | 12.11 | 8.53–15.69 |
| | > 3 | 80 | 19.90 | 43.75 | 32.83–54.65 |

Considering the other variables resulting from the hospital stay, most patients (n = 310; 77.11%) underwent surgical procedures, and the most frequently performed one was osteosynthesis (n = 173; 13.87%). Nonetheless, the highest mortality was found among patients who did not undergo surgery (n = 92; 38.04%). Most patients did not present infection during hospital stay (n = 267; 66%) and stayed less than three days in the ICU (n = 322; 80,1%). The number of patients diagnosed with PTE was low (n = 18; 4.48%).

Despite this, the highest incidence of mortality occurred among patients who had some type of infection (n = 135; 34%). Considering the affected topographic sites, 86 patients with respiratory infection (21.39%), the outcome death occurred in 69.77% (60.02–79.5), followed

by those with urinary infection (n. = 64; 15.92%), with 40.62% (28.54–52.71) mortality. The same occurred with patients who remained in the ICU for more than 3 days (43.75%; 32.83–54.65) and had pulmonary embolism (n = 18; 38.48–83.73).

## Mortality and risk variables

In the bivariate analysis, a statistically significant association was observed between mortality and the following variables: age groups 80–89 years (RR = 3.52; 95% CI 2.00–6.17) and over 90 years (RR = 6.13; 95% CI 3.45–10.90); hemoglobin ≤ 10 (RR = 1.85; 95% CI 1.21–2.81); neurological disorders (RR = 1.60; 9% CI 1.06–2.44), COPD (RR = 3.04; 95% CI 1.97–4.69), and high surgical risk (RR = 4.43; 95% CI 2.84–6.93).

Among the in-hospital variables, non-surgical interventions (RR = 3.47; 95% CI 2.01–5.99); respiratory infection (RR = 15.75; 95% CI 9.26–26.77), urinary tract infection (RR = 2.86; 95% CI 1.93–4.25), or other infections (RR = 2.52; 95% CI 1.39–4.58); PTE (RR = 3.72; 95% CI 2.42–5.74); and a stay in ICU for more than three days (RR = 3.61; 95% CI 2.46–5.31) were also significantly associated with the mortality outcome (Table 3).

Using the Poisson regression model with robust variance, in the first stage of the hierarchical model (epidemiological block) the sex and age variables were included. The age groups 80–89 and over 90 years showed a significant association with mortality. Therefore, the age variable was maintained for the next block analysis.

In the second stage, age was included along with the block of in-hospital variables, and only the variables hemoglobin ≤ 10 (PR = 1.54; 95% CI 1.03–2.29), COPD (PR = 2.39; 95% CI 1.52–3.78), and high surgical risk (PR = 3.18; 95% CI 2.00–5.16) showed a significant association with mortality, even after adjustment. These variables were maintained for the next block analysis.

In the last stage, in addition to the variables age, hemoglobin, COPD, and high surgical risk, the postoperative variables block was included. After adjustment for possible confounders, the variables respiratory infection (PR = 7.27; 95% CI 3.98–13.26), urinary tract infection (PR = 2.04; 95% CI 1.44–2.89), other infections (PR = 1.98; 95% CI 1.09–3.62), and PTE (PR = 1.98; 95% CI 1.11–3.52), all in relation to the bivariate analysis, showed a significant association with mortality.

## Mortality and time variables

Considering the entire sample, patients who died had a longer hospital stay (31.28 ± 23.24); however, when comparing each variable, related to time with mortality outcome (yes or no), the postoperative period (21.05 ± 21.71) and longer stay in ICU (10.65 ± 16.62) presented a significant association with mortality (Table 4).

When comparing and analyzing the time variables among the groups, the mean total length of hospital stay was significantly longer among patients who died in both groups and in both those undergoing arthroplasty (p = 0.0015) and osteosynthesis (p = 0.0005).

The same was identified for the mean postoperative period, which was significantly longer among patients who died in both groups and both among those undergoing arthroplasty (p < 0.0001) and osteosynthesis (p < 0.0001).

In turn, the mean preoperative period did not show any significant difference between the surviving and mortality groups and both in those undergoing arthroplasty (p = 0.6614) and osteosynthesis (p = 0.4211), as shown in Table 5.

The mean stay in ICU in days was significantly longer among patients who died in all groups, both for those undergoing arthroplasty (p < 0.0001) and osteosynthesis (p < 0.0001) and for those who did not undergo surgery (p = 0.0002).

**Table 3. Relative risk and prevalence ratio using the Poisson regression model with robust variance and its respective 95% confidence interval.**

| Variables | Relative Risk (RR) | | Relative Risk (RR) Adjusted* | |
|---|---|---|---|---|
| | RR (95% IC) | p-value | RR (95% IC) | p-value |
| **Block 1 –Demographic** | | | | |
| **Sex** | | 0.4394 | | 0.5128 |
| Women | 1 | - | 1 | - |
| Men | 1.18 (0.78–1.79) | 0.4394 | 1.14 (0.77–1.69) | 0.5128 |
| **Age** | | < 0.0001 | | < 00001 |
| 60–79 | 1 | - | 1 | - |
| 80–89 | 3.52 (2.00–6.17) | < 0.0001 | 350 (1.99–6.15) | < 0.0001 |
| > 90 | 6.13 (3.45–10.90) | < 0.0001 | 6.11 (3.44–10.87) | < 0.0001 |
| **Block 2 –Preoperative** | | | | |
| **Hemoglobin** | | | | 0.0334 |
| ≤ 10 | 1.85 (1.21–2.81) | 0.0041 | 1.54 (1.03–2.29) | 0.0334 |
| > 10 | 1 | - | 1 | - |
| **SAH** | | 0.0901 | | 0.2475 |
| No | 1 | - | 1 | - |
| Yes | 1.64 (0.92–2.92) | 0.0901 | 1.39 (0.79–2.45) | 0.2475 |
| **DM** | | 0.1412 | | 0.6915 |
| No | 1 | - | 1 | - |
| Yes | 1.36 (0.90–2.06) | 0.1412 | 0.92 (0.63–1.36) | 0.6915 |
| **Neurological disorders** | | 0.0267 | | 0.5050 |
| No | 1 | - | 1 | - |
| Yes | 1.60 (1.06–2.44) | 0.0267 | 1.14 (0.77–1.69) | 0.5050 |
| **COPD** | | < 0.0001 | | 0.0002 |
| No | 1 | - | 1 | - |
| Yes | 3.04 (1.97–4.69) | < 0.0001 | 2.39 (1.52–3.78) | 0.0002 |
| **Surgical risk** | | < 0.0001 | | < 0.0001 |
| Low to moderate | 1 | - | 1 | - |
| High | 4.43 (2.84–693) | <0.0001 | 3.18 (2.00–5.06) | < 0.0001 |
| **Femur fracture** | | 0.8222 | | 0.1681 |
| Intracapsular | 1 | - | 1 | - |
| Extracapsular | 0.95 (0.63–1.44) | 0.8222 | 1.32 (0.89–1.97) | 0.1681 |
| **Block 3 –Postoperative** | | | | |
| **Type of surgery** | | < 0.0001 | | 0.0003 |
| Non-surgical | 3.47 (2.01–5.99) | < 0.0001 | 2.04 (1.31–3.16) | 0.0003 |
| Osteosynthesis | 1.27 (0.69–2.32) | 0.4431 | 1.03 (0.64–1.64) | 0.9108 |
| Arthroplasty | 1 | - | | |
| **Respiratory infection** | | < 0.0001 | | < 0.0001 |
| No | 1 | - | 1 | - |
| Yes | 15.75 (9.26–26.77) | < 0.0001 | 7.27 (3.98–3.26) | < 0.0001 |
| **Urinary tract infection** | | < 0.0001 | | < 0.0001 |
| No | 1 | - | 1 | - |
| Yes | 2.86 (1.93–4.25) | < 0.0001 | 2.04 (1.44–2.89) | < 0.0001 |
| **Other infection** | | 0.0024 | | 0.0251 |
| No | 1 | - | 1 | - |
| Yes | 2.52 (1.39–4.58) | 0.0024 | 1,98 (1.09–3.62) | 0.0251 |
| **PTE** | | < 0.0001 | | 0.0202 |
| No | 1 | - | 1 | - |

*(Continued)*

**Table 3.** (Continued)

| Variables | Relative Risk (RR) | | Relative Risk (RR) Adjusted* | |
|---|---|---|---|---|
| | RR (95% IC) | p-value | RR (95% IC) | p-value |
| Yes | 3.72 (2.42–5.74) | < 0.0001 | 1.98 (1.11–3.52) | 0.0202 |
| **Days in ICU** | | < 0.0001 | | 0.1683 |
| ≤ 3 | 1 | - | 1 | - |
| > 3 | 3.61 (2.46–5.31) | < 0.0001 | 1.27 (0.90–1.80) | 0.1683 |

Finally, to estimate the number of postoperative days, regardless of the surgical technique employed, after which there was a statistical significance concerning the incidence of mortality in these patients, an analysis was performed using the ROC curve (Fig 1). After 22 days (95% CI 0.4998–0.6634) of hospital stay, there was a correlation with mortality in the entire sample. The same applied six days after surgery, regardless of the surgical technique employed (95% CI 0.6884–0.8959) (Figs 2 and 3) (S1 Database).

## Discussion

Hip fractures are strongly associated with increased mortality rates in different studies worldwide [21]. In recent years, predictive models for mortality risk after hip fractures have been developed to identify patients at a higher risk and propose intervention strategies to improve the outcomes of hip fractures [9, 22–25].

The mean hospital mortality rate considering the nine years of study was 18.4%, and the variables with a significant association with the mortality outcome in elderly patients hospitalized for osteoporotic hip fractures were: 1) demographic factors: age over 90 years; 2) factors associated with clinical conditions and comorbidities: high preoperative cardiovascular risk, hemoglobin ≤ 10, chronic obstructive pulmonary disease (COPD); 3) hospital factors: respiratory infection, urinary tract infection, and other infections, pulmonary thromboembolism (PTE).

The challenge of establishing a pattern among the several variables associated with worsening and to mortality outcomes [26] is related to the fact that these associations, as well as the mortality rate assessments reported in different studies involving different populations and methodologies, also present regional variations [17, 18, 27–29].

In our study, by examining the variables selected for analysis, we can try to identify the sample behavior synthesizing the existing relationships. However, as the study purpose was not to compare the independent variables with each other, but to compare their association with the mortality outcome, we cannot attest, but only discuss the existence of a causal relationship among them.

**Table 4. Total mean of hospital stay, preoperative period, postoperative period and days in ICU when comparing the surviving and mortality groups.**

| Variables | Patients (n) | Mean (days) | SD | Mortality | | p-value* |
|---|---|---|---|---|---|---|
| | | | | Yes (days) | No (days) | |
| **Total mean of hospital stay (days)** | 402 | 25.39 | 18.91 | 31.28±23.24 | 24.06±17.56 | 0.0283 |
| **Preoperative period (days)** | 311 | 19.22 | 14.44 | 18.62±12.07 | 19.30±14.76 | 0.8191 |
| **Postoperative period (days)** | 308 | 7.23 | 10.47 | 21.05±21.71 | 5.23±5.20 | <0.0001 |
| **Stay in ICU (days)** | 402 | 3.71 | 8.82 | 10.65±16.62 | 2.15±4.51 | <0.0001 |

* to calculate p-value the Mann-Whitney test was used

**Table 5. Total mean of hospital stay, preoperative period, postoperative period and days in ICU when comparing the surviving and mortality groups, regarding the surgery performed.**

|  | Arthroplasty | | | Osteosynthesis | | | Non-surgical | | |
|---|---|---|---|---|---|---|---|---|---|
| **Variables**# | Mortality | | | Mortality | | | Mortality | | |
|  | **No** | **Yes** | **p-value**\* | **No** | **Yes** | **p-value**\* | **No** | **Yes** | **p-value**\* |
| **Total mean of hospital stay** | 29.04±24.09 | 48.67±28.26 | 0.0015 | 20.82±10.54 | 33.29±19.52 | 0.0005 | 21.86±12.93 | 22.46±18.88 | 0.4914 |
| **Preoperative period** | 22.57±18.79 | 19.73±14.85 | 0.6614 | 16.75±9.68 | 17.92±10.26 | 0.4221 | - | - | - |
| **Postoperative period** | 5.91±5.86 | 29.27±26.57 | <0.0001 | 4.42±3.11 | 15.92±16.64 | <0.0001 | - | - | - |
| **Stay in ICU** | 2.77±5.08 | 18.33±21.90 | <0.0001 | 2.24±4.29 | 12.88±16.78 | <0.0001 | 0.58±3.30 | 5.83±12.33 | 0.0002 |

When proposing this study, the authors gathered from the observation of real-life outcomes—discharge or death, the several hypotheses that originated the research questions, either from the health conditions of the population assisted, or the course of fracture treatment from admission to the intervention, and studied a method of data collection and analysis development, regarding the variables of interest, that demonstrated significance for the construction of evidence.

Considering the descriptive analysis, demographic factors show a higher incidence of hospital admission in elderly women, which is meets the profile of patients affected by osteoporotic femur fractures after a fall from standing height [1]. However, in the frequency analysis, the incidence of male mortality in the general sample was higher. Studies show that the excess mortality in men remains high for up to 20 years after the fracture [30], but the causes for this difference in mortality in absolute terms are not fully explained yet [31, 32]. A comparative study between sexes attributed the difference observed to the relationship between deaths and infections (pneumonia, influenza, and septicemia) [33].

Considering the increase in life expectancy worldwide, especially in developing countries [34], the association among longevity, osteoporotic hip fractures, and mortality becomes relevant [29], raising the question of how these injuries affect health systems [35]. The age group over 90 years, which was identified by the hierarchical model of multiple regression as a factor associated with a six-fold higher RR for death [RR 6.11 (3.44–10.87)], relates the factors attributed to the senescence of this age group [36] both with bone degeneration, which accounts for

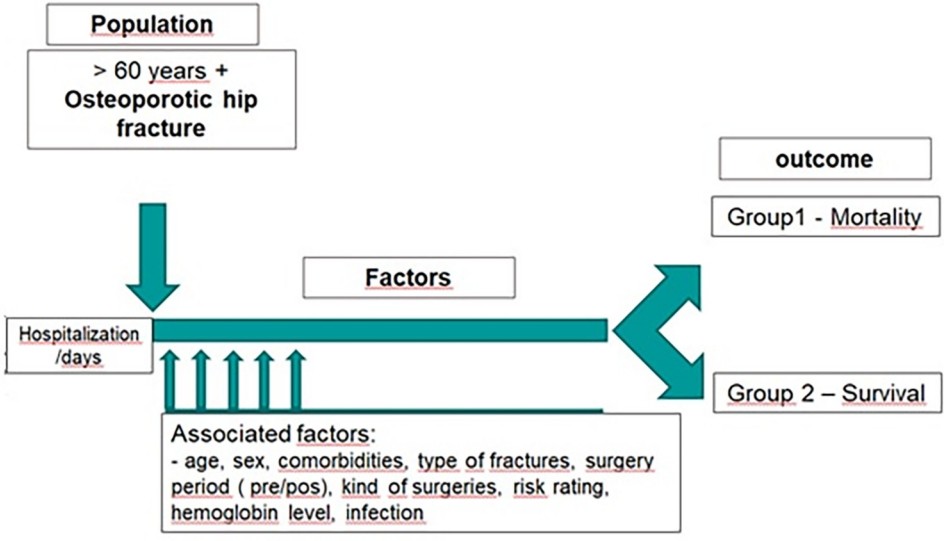

**Fig 1. Process model of data extraction and patients' follow-up during hospitalization.**

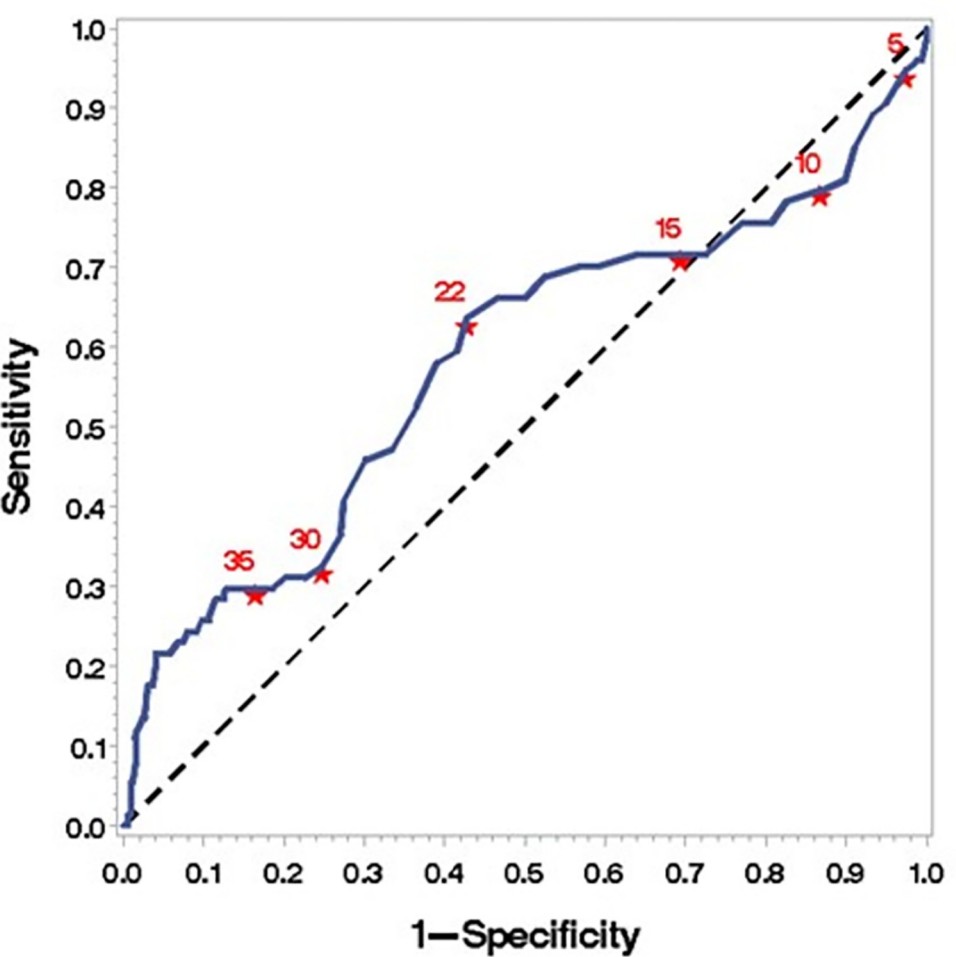

**Fig 2. Cutoff point for mortality from the 22nd day of the total length of hospital stay.** ROC (curve area 0.5816; 95% IC 0.4998–0.6634).

the highest incidence of fracture mechanisms [37], and with a higher risk of fracture complications [38].

Other factors in our study can also contribute to the regional population analysis and reflect the health conditions related to aging. Almost 90% of patients over 60 years of age presented with some comorbidity at hospital admission. The literature already described that the comorbidities identified at admission are related to the mortality outcome in patients with hip fracture in the short and long term [2]. We did not use quantitative comorbidity rates in our study but choose to present them as individual variables. It seems desirable to efficiently summarize one or more comorbidities in a single score [39]; however, the purpose of this work was to obtain a greater nosological perspective of each variable behavior separately in relation to mortality and their relationship with osteoporosis as an underlying disease [40, 41].

Among the comorbidities grouped as preoperative variables, the Poisson regression model with robust variance related COPD with a significant 2-fold higher mortality risk compared to non-COPD patients. A study also compared the mortality between COPD and non-COPD patients and revealed that COPD was an independent mortality factor over a minimum follow-up period of one year and that the disease severity in patients with hip fracture was also a risk factor for mortality for six months to one year [42].

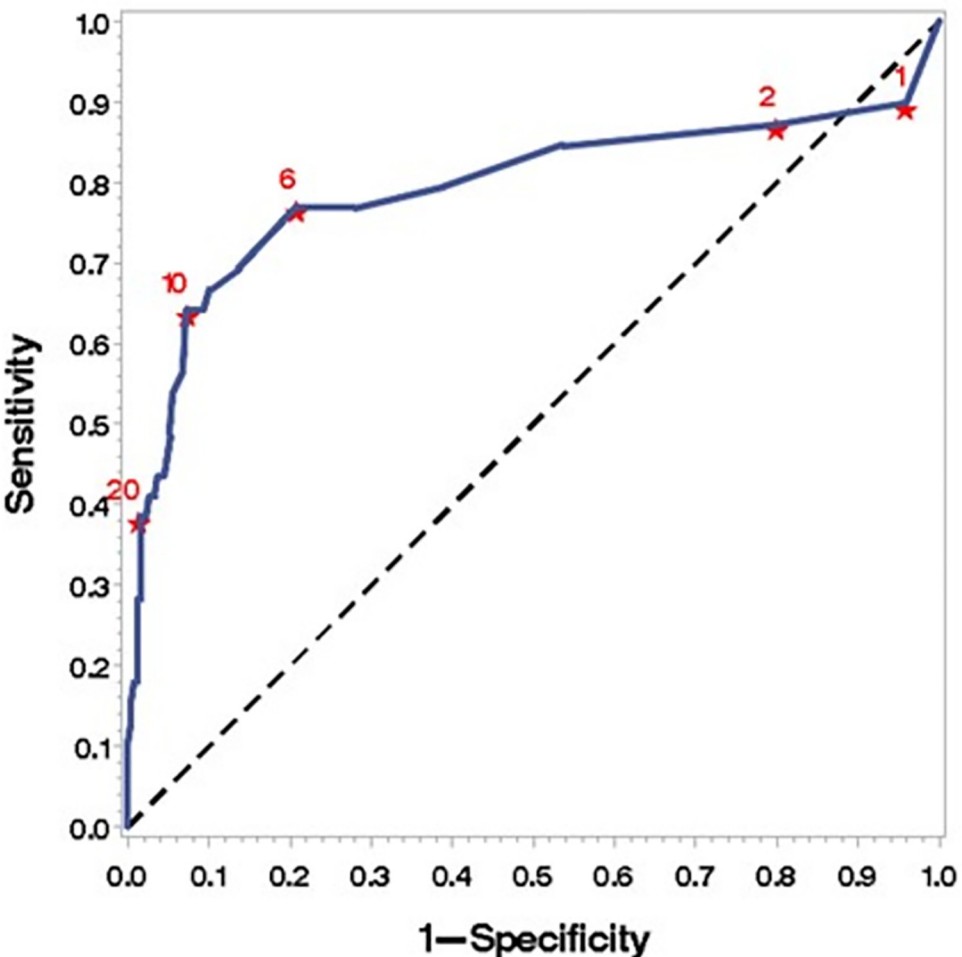

**Fig 3. Cutoff point for mortality from the sixth day on for the postoperative period.** ROC (curve area 0.7922; 95% IC 0.6884–0.8959).

Hemoglobin was also selected to analyze the impact of blood levels below 10 g/dL on the mortality outcome. Elderly patients with hip fractures are known to have a high risk of perioperative anemia due to blood loss related to the fractures and/or surgery [43]. Besides, studies have shown that patients who need blood transfusion require a longer hospital stay [44]. In our study, even after adjusting the model variables, there was a significant 1.5-fold higher risk of mortality in patients with levels above 10 g/dL, making it an important variable to be analyzed.

Regarding the variable DM, it should be highlighted that, despite not being included in the multiple regression model as a variable directly associated with the mortality risk, the group of patients with DM had higher mortality in the frequency description. Studies already related mortality after hip fractures with DM [45], which represents an increased risk factor in case of fractures due to fragility that seems to be independent of bone mineral density [46].

Preoperative cardiac risk was analyzed using the Detsky Modified Cardiac Risk Index, 1986 [47], and patients classified as high surgical risk (RR 10.6) were significantly associated with the highest risk of death in our study. This assessment can be used to estimate possible risks resulting from the surgical procedure in each patient and, if possible, conducts to minimize

these risks. This estimate is essential to provide the surgeon/team and patient/family with information that must be taken into account when comparing the procedure's possible benefits and harms in each case [48].

The association between atherosclerotic cardiovascular diseases and osteoporosis emphasizes epidemiological and physiopathogenic similarities in arterial wall calcification and osteogenesis considering the low bone mass, osteoporosis fractures, vascular calcification, extension of coronary and abdominal aorta injury, and cardiovascular mortality, regardless of age. Osteopenia and osteoporosis in the femoral neck were associated with a higher risk of severe coronary lesions [49, 50], and the reverse [50].

However, the Detsky Modified Cardiac Risk Index, 1986 [47] does not consider the SAH, present in 75% of patients in our study, in its score. Although this variable was not associated with a direct risk of death, patients with SAH had a higher mortality.

The high prevalence of cardiovascular diseases found in the elderly population in Brazil [51] raises two issues: 1) the established relationship between cardiovascular diseases and osteoporosis [52], and 2) SAH as an independent risk factor for increased mortality [53] to be considered, requiring evaluation regarding its prevention and associated risk factors.

Nonetheless, infection was the prevailing factor for in-hospital mortality in elderly patients with femoral fractures in this study, as per the hierarchical model of multiple regression after adjustment for possible confounders. Patients with pulmonary infection had a seven-fold higher risk of death than patients who did not have this condition.

Previous studies demonstrate pulmonary infection during hospital stay as a risk factor for death associated with one of the most common complications [53, 54], also representing an independent risk for early readmission after hip fracture surgery [55].

Evidence shows a relationship between trauma and an age-related decline in the elderly health, affecting the neutrophils function and reducing their immune response to bacteria [56]. Additionally, in patients who already have a higher incidence of pre-existing cardiorespiratory disease and reduced mobility after a hip fracture, there is an increased risk for pneumonia [55]. These were characteristics presented by the patients included in this study.

Other infections were also associated with the mortality risk in this population, such as urinary tract infections, which complies with findings of other studies that made this correlation with other complications similar to surgical site infection [18], such as progression to sepsis and extended hospital stay [57].

Regarding the hip fracture trauma mechanism pattern, these are classified by their site as an evidence of prognostic implications [58]. Surgery should be the most appropriate option for most patients [59], as it is associated with a shorter hospital stay and better rehabilitation [60]. Non-surgical interventions are reserved for patients with a severe debilitation, unstable patients with incurable severe diseases, or patients with terminal illnesses in the final stages of life [61]. Stable impacted fractures can also be considered for non-surgical treatment [62]. However, with conservative treatment, rehabilitation will probably be slower and limb deformity, more common [63].

Further studies are required to allow a shared decision-making, and questions about the pre-fracture quality of life and future perspectives should be asked before considering different treatment options to assess which one is advisable in frail and high-risk elderly patients, considering that most patients with hip fracture also have advanced comorbidities [8].

Finally, we analyzed the time variables with and without surgery comparing the techniques employed, which did not differ in relation to the risk of death. However, it was found that an increased total length of hospital stay, prolonged postoperative period, and longer ICU stay are associated with a higher mortality, which also applies to patients who have not undergone surgical treatment.

Analysis limitations should be considered for associations related to increased postoperative mortality, such as the performance or not of early mobilization after surgery. However, PTE occurrence and its relationship with mortality can suggest a deficiency in postoperative rehabilitation protocols.

There are also considerations regarding the intervention period as a limiting factor. In our study, the preoperative period did not significantly influence mortality, in contrast to studies that recommend early intervention [10]. However, due to fewer early intervention, the analysis in this group may have been insufficient for the effect of an association with significance.

On the other hand, chronic diseases can contribute to the hypothesis of aggravation or worsening, leading to the fall from their height, and therefore the health status at admission that does not allow early surgical intervention, as well as contributing further complications after surgical stress.

Despite the Brazilian Unified Health System being universal and of unrestricted access, health interventions may be conditioned to the availability of resources, equipment, and management issues of the public system, which is a reason to question also whether there were limitations in the management of the pathology, or if only the patient's previous health conditions at admission, such as multiple comorbidities, made surgery unfeasible if there was no early intervention, considering the need for a period of clinical stability for the patient to be suitable for surgery.

A study on the length of hospital stay for the pathology in public hospitals in Brazil, in a 10-year historical analysis, considering 480,652 hospitalizations, confirms the longest mean length of hospital stay among Brazilian capitals, in the Federal District (18.7 days), the geographical region where the research was developed [64].

Therefore, this study showed the importance of comprehensive treatment in the fracture approach, from the clinical condition of the patients to treatment time, including rehabilitation and clinical care after the intervention. We suppose that often after discharge, complications are not reported (underreporting). The extended in-hospital observation of this study, on the other hand, was able to capture.

## Conclusions

Although the number of hospitalizations was higher in elderly women with multiple comorbidities, the male mortality was higher. Among the comorbidities studied, mortality rates were higher in patients with diabetes, hypertension, COPD, and risk attributed to a high preoperative risk score and lower serum hemoglobin levels. The impact of these comorbidities, acting as "correlation variables," on the in-hospital variables should be considered, since the highest mortality occurs in patients who did not undergo surgery or who were hospitalized for longer periods, including ICU stay, therefore attributing a greater severity in the condition of these patients upon admission. Among the in-hospital variables, infections were the most prevalent factor associated with in-hospital mortality, especially respiratory ones. The aim of the study was not to compare independent variables with each other, but the association of comorbidities leading to the development of infections directly linked to mortality is clear. These factors emphasize the attention required by individual perspectives on healthy aging promotion and by politics and programs ensuring access to the health system, preventing comorbidities and falls and creating strategies for early risk assessment, and causal effect estimates, which help us understand the impact of different treatment decisions made in that population, to prevent the mortality outcome.

Faced with the evidence generated by the scientific community worldwide on mortality in the aspect of femur fracture in the elderly, the study sought dialogue between the results of this

study and the international bibliographic references. The point discussed in the review regarding the low early intervention in this study was also honestly mentioned in the text.

The study aimed, within the methods' limits, to separate the variables related to the individual and the variables related to the health system in the care of these patients, not to generate confusion but to clarify to the readers the main aspects that would be associated to the mortality outcome.

## Supporting information

**S1 Database. Study's database.**
(XLSX)

**S1 File. Statistical reporting.**
(DOCX)

## Author Contributions

**Conceptualization:** Viviane Cristina Uliana Peterle, Maria Rita Carvalho Garbi Novaes.

**Data curation:** Viviane Cristina Uliana Peterle, Maria Rita Carvalho Garbi Novaes, João Carlos Geber Júnior, Rodrigo Tinôco Magalhães Cavalcante, Jurandi Barrozo da Silva Junior, Ray Costa Portela.

**Formal analysis:** Viviane Cristina Uliana Peterle, Maria Rita Carvalho Garbi Novaes, Paulo Emiliano Bezerra Junior, João Carlos Geber Júnior.

**Funding acquisition:** Maria Rita Carvalho Garbi Novaes.

**Investigation:** Viviane Cristina Uliana Peterle, Maria Rita Carvalho Garbi Novaes, Paulo Emiliano Bezerra Junior, Rodrigo Tinôco Magalhães Cavalcante, Jurandi Barrozo da Silva Junior, Ray Costa Portela, Ana Patricia de Paula.

**Methodology:** Viviane Cristina Uliana Peterle, Maria Rita Carvalho Garbi Novaes, Paulo Emiliano Bezerra Junior, João Carlos Geber Júnior.

**Project administration:** Viviane Cristina Uliana Peterle, Maria Rita Carvalho Garbi Novaes, Paulo Emiliano Bezerra Junior, João Carlos Geber Júnior.

**Resources:** Viviane Cristina Uliana Peterle, Maria Rita Carvalho Garbi Novaes, Paulo Emiliano Bezerra Junior, João Carlos Geber Júnior.

**Software:** Maria Rita Carvalho Garbi Novaes, Paulo Emiliano Bezerra Junior, João Carlos Geber Júnior.

**Supervision:** Viviane Cristina Uliana Peterle, Maria Rita Carvalho Garbi Novaes.

**Validation:** Viviane Cristina Uliana Peterle, Maria Rita Carvalho Garbi Novaes, Paulo Emiliano Bezerra Junior, João Carlos Geber Júnior.

**Visualization:** Viviane Cristina Uliana Peterle, Maria Rita Carvalho Garbi Novaes, Paulo Emiliano Bezerra Junior, João Carlos Geber Júnior, Rodrigo Tinôco Magalhães Cavalcante, Jurandi Barrozo da Silva Junior, Ray Costa Portela.

**Writing – original draft:** Viviane Cristina Uliana Peterle, Maria Rita Carvalho Garbi Novaes, Paulo Emiliano Bezerra Junior, João Carlos Geber Júnior.

**Writing – review & editing:** Viviane Cristina Uliana Peterle, Maria Rita Carvalho Garbi Novaes, Ana Patricia de Paula.

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
