## [Decision Letter · Decision Letter 0]

22 Mar 2022

PONE-D-22-03245Osteoporotic hip fracture – comorbidities and factors associated with in-hospital mortality in the elderly: a nine-year cohort study in BrazilPLOS ONE

Dear Dr. Peterle,

Thank you for submitting your manuscript to PLOS ONE. After careful consideration, we feel that it has merit but does not fully meet PLOS ONE’s publication criteria as it currently stands. Therefore, we invite you to submit a revised version of the manuscript that addresses the points raised during the review process.

Both reviewers identify important details that have been omitted from your MS.  Complete description of the methods and population are necessary.

We look forward to receiving your revised manuscript.

Kind regards,

Robert Daniel Blank, MD, PhD

Academic Editor

PLOS ONE

Journal Requirements:

Reviewers' comments:

Reviewer's Responses to Questions

**Comments to the Author**

1. Is the manuscript technically sound, and do the data support the conclusions?

Reviewer #1: Partly

Reviewer #2: Yes

2. Has the statistical analysis been performed appropriately and rigorously? 

Reviewer #1: Yes

Reviewer #2: Yes

3. Have the authors made all data underlying the findings in their manuscript fully available?

Reviewer #1: Yes

Reviewer #2: No

4. Is the manuscript presented in an intelligible fashion and written in standard English?

Reviewer #1: Yes

Reviewer #2: Yes

5. Review Comments to the Author

Reviewer #1: the authors have captured 400 hip fracture patients over 9 years and characterized the factors that influenced mortality. Their results are consistent with a number of other similar publications. This is unique in that it addressed Brazil medicine.

There are some key concerns with the patient population. This study enters less than one patient per week for hip fractures. The usual general hospital that handles hip fractures sees one per day. Were the 400 patients all the hip fractures at that hospital. If no how were they chosen and was there potential bias. If these are all the patients then the hospital is weak in hip fracture experience and why so few? Where did these patients come from?

Also of concern is the long hospital stay. Around the world most hip fracture patients leave after 4-5 days going either home or to a rehab hospital. How do the authors explain this markedly long hospital stay. Are the patients both acute and then subacute??

Time to surgery has consistently been demonstrated to influence mortality. Why not in this group?? It is recommended that the patients should undergo surgery ideally within 24 hours of admission and certainly within the first 48 hours. How did this hospital rate and did the patients who were operated within 24 hours do compared to the others?

Most of the medical issues are covered by the Charlson index and most of the OR risk is covered by the ASA score. Please provide them for this patient population

this study has a very high mortality rate. give us the 3-day rate rather than the 22-day rate.

Reviewer #2: This is a retrospective multivariate analysis from Brazil of 402 patients with hip fractures over eight years to examine factors associated with in hospital mortality. The in-hospital mortality was 18% and not surprisingly older patients, those with comorbidities, those selected for non-operative treatment, and who develop infections were at risk for higher mortality.

The study was very comprehensive and its analysis of important variables and include some relatively novel methods of stratifying risk such as cardiac, pulmonary embolism etc. with quantitative scales. The Multi variable analysis was extremely well done and eloquently explained in the results. The results and conclusions are justified by their data and analytic methods. The manuscript was extremely well written and a pleasure to read.

A number weaknesses need to be taken into consideration when interpreting those data. First I do not think the ROC analysis on length of stay is compelling or generalizable and depends on local conditions. I do not believe it is useful and would be best deleted from the study. The selection bias regarding surgical versus non-surgical treatment is problematic and clearly results in the patients treated nonoperatively having higher mortality. One approach might be the patient altogether and only discussing hospital mortality of the operative patients or do this separately. The follow up is very short and only regarding inpatient mortality which may not generalize well to the rest of the world given the prolong length of stay seen in Brazil.

101 It seems like the aims should be stated in the introduction not the conclusion. Also, in the abstract you did not comment on the results regarding infections.

145 What was the time length of follow-up? Is this only in hospital mortality?

149 you have a very good list of comorbidities however the cognitive state of the patient either acute or chronic are important predictors of outcome, risk of complications, mortality, and are perhaps the most important aspect regarding surgical decision making.

199 the ROC analysis regarding length of stay is very dependent on the cultural and healthcare system utilized in Brazil and does not apply elsewhere. Also, significant bias in the patient to die early would have short length of stay but those who have prolong illness as long as he stays would have prolonged length of stay this would seem to confound this analysis. Please explain what postoperative time period means.

305 Please explain what groups you were referring to.

324 Need to define whether this is total hip arthroplasty or hemi arthroplasty.

325 One of the major weaknesses of the study is that you do not specify indications for osteosynthesis, arthroplasty or non-operative treatment. These are largely dependent fracture type as osteosynthesis would typically use for intertrochanteric, subtrochanteric ,and some femoral neck fractures where is hemiarthroplasty for displaced femoral neck fractures. Since these cannot be used interchangeably it makes no sense comparing these surgeries. Further non operative care is typically selected for those with greater trochanteric fractures or in patients too sick to have surgery or who have a very short life expectancy. Thus, significant selection bias is present which accounts for the higher mortality.

463 I think you need to significantly expand study limitations particular problem of selection bias, unknown indications for surgery, not including cognitive state patient, only including in-hospital mortality.

6. PLOS authors have the option to publish the peer review history of their article (what does this mean?). If published, this will include your full peer review and any attached files.

Reviewer #1: No

Reviewer #2: No

---

## [Author Response · Author response to Decision Letter 0]

6 May 2022

Dear Editor,

Thank you for considering the paper " Osteoporotic hip fracture – comorbidities and factors associated with in-hospital mortality in the elderly: a nine-year cohort study in Brazil" by Peterle et al., for publication in the PlosOne Journal and allowing us to re-submit this revised manuscript as a letter to the editor.

The authors are grateful to the reviewers for the careful appraisal, positive comments, and helpful criticisms. Suggested changes have been addressed. We believe that the quality of the manuscript has been improved and hope that it now meets the quality required for publication in the PlosOne Journal.

A detailed point-by-point response is given below.

We are looking forward to your decision regarding the suitability of the revised version of this paper for the Journal.

Thank you very much in advance,

Viviane Cristina Uliana Peterle*, MD, PhD

*Corresponding Author 

E-mail: vivianepeterle@hotmail.com

Orcid: https://orcid.org/0000-0003-1693-242X

Afiliation 1: Escola Superior de Ciências da Saúde (Escs/Fepecs), Brasília, DF, Brazil Afiliation 2: Universidade de Brasilia, Brasília, DF, Brazil

---

## [Decision Letter · Decision Letter 1]

30 May 2022

PONE-D-22-03245R1Osteoporotic hip fracture – comorbidities and factors associated with in-hospital mortality in the elderly: a nine-year cohort study in BrazilPLOS ONE

Dear Dr. Peterle,

Thank you for submitting your manuscript to PLOS ONE. After careful consideration, we feel that it has merit but does not fully meet PLOS ONE’s publication criteria as it currently stands. Therefore, we invite you to submit a revised version of the manuscript that addresses the points raised during the review process.

Reviewer 1 has asked that you add a brief discussion about the limitations of care in your setting, relative to best practices in wealthy countries.  This is a good idea and should be done.  

We look forward to receiving your revised manuscript.

Kind regards,

Robert Daniel Blank, MD, PhD

Academic Editor

PLOS ONE

Journal Requirements:

Reviewers' comments:

Reviewer's Responses to Questions

**Comments to the Author**

1. If the authors have adequately addressed your comments raised in a previous round of review and you feel that this manuscript is now acceptable for publication, you may indicate that here to bypass the “Comments to the Author” section, enter your conflict of interest statement in the “Confidential to Editor” section, and submit your "Accept" recommendation.

Reviewer #1: All comments have been addressed

Reviewer #2: All comments have been addressed

2. Is the manuscript technically sound, and do the data support the conclusions?

Reviewer #1: Yes

Reviewer #2: Yes

3. Has the statistical analysis been performed appropriately and rigorously? 

Reviewer #1: Yes

Reviewer #2: Yes

4. Have the authors made all data underlying the findings in their manuscript fully available?

Reviewer #1: Yes

Reviewer #2: Yes

5. Is the manuscript presented in an intelligible fashion and written in standard English?

Reviewer #1: Yes

Reviewer #2: Yes

6. Review Comments to the Author

Reviewer #1: The authors have stressed the underdeveloped nature of their country and feel that they should not be held to the highest level of care. There is a large number of patients that are not operated in a timely fashion. If we accept this concept of a developing nation, then the article is clear and well defined within their capacity. The only point would be in the abstract to clearly identify this limitation to the readers.

Reviewer #2: (No Response)

7. PLOS authors have the option to publish the peer review history of their article (what does this mean?). If published, this will include your full peer review and any attached files.

Reviewer #1: No

Reviewer #2: No

---

## [Author Response · Author response to Decision Letter 1]

4 Jul 2022

Dear Editor,

Thank you for considering the paper "Osteoporotic hip fracture – comorbidities and factors associated with in-hospital mortality in the elderly: a nine-year cohort study in Brazil" by Peterle et al., for publication in the PlosOne Journal and allowing us to re-submit this revised manuscript as a letter to the editor.

The authors are grateful to the reviewers for the careful appraisal, positive comments, and helpful criticisms. Suggested changes have been addressed. We believe that the quality of the manuscript has been improved and hope that it now meets the quality required for publication in the PlosOne Journal.

A detailed point-by-point response is given below.

We are looking forward to your decision regarding the suitability of the revised version of this paper for the Journal.

Thank you very much in advance,

Viviane Cristina Uliana Peterle*, MD, PhD

*Corresponding Author 

E-mail: vivianepeterle@hotmail.com

Orcid: https://orcid.org/0000-0003-1693-242X

Afiliation 1: Escola Superior de Ciências da Saúde (Escs/Fepecs), Brasília, DF, Brazil Afiliation 2: Universidade de Brasilia, Brasília, DF, Brazil

---

## [Decision Letter · Decision Letter 2]

12 Jul 2022

Osteoporotic hip fracture – comorbidities and factors associated with in-hospital mortality in the elderly: a nine-year cohort study in Brazil

PONE-D-22-03245R2

Dear Dr. Peterle,

We’re pleased to inform you that your manuscript has been judged scientifically suitable for publication and will be formally accepted for publication once it meets all outstanding technical requirements.

Kind regards,

Robert Daniel Blank, MD, PhD

Academic Editor

PLOS ONE

Additional Editor Comments (optional):

Reviewers' comments:

Reviewer's Responses to Questions

**Comments to the Author**

1. If the authors have adequately addressed your comments raised in a previous round of review and you feel that this manuscript is now acceptable for publication, you may indicate that here to bypass the “Comments to the Author” section, enter your conflict of interest statement in the “Confidential to Editor” section, and submit your "Accept" recommendation.

Reviewer #1: All comments have been addressed

Reviewer #2: All comments have been addressed

2. Is the manuscript technically sound, and do the data support the conclusions?

Reviewer #1: Yes

Reviewer #2: Yes

3. Has the statistical analysis been performed appropriately and rigorously? 

Reviewer #1: Yes

Reviewer #2: Yes

4. Have the authors made all data underlying the findings in their manuscript fully available?

Reviewer #1: Yes

Reviewer #2: Yes

5. Is the manuscript presented in an intelligible fashion and written in standard English?

Reviewer #1: Yes

Reviewer #2: Yes

6. Review Comments to the Author

Reviewer #1: The minor issues in the manuscript have been corrected except stating in the abstract that this is a developing country.

Reviewer #2: (No Response)

7. PLOS authors have the option to publish the peer review history of their article (what does this mean?). If published, this will include your full peer review and any attached files.

Reviewer #1: No

Reviewer #2: No

---

## [Editor Report · Acceptance letter]

4 Aug 2022

PONE-D-22-03245R2 

Osteoporotic hip fracture – comorbidities and factors associated with in-hospital mortality in the elderly: a nine-year cohort study in Brazil 

Dear Dr. Peterle:

I'm pleased to inform you that your manuscript has been deemed suitable for publication in PLOS ONE. Congratulations! Your manuscript is now with our production department. 

Kind regards, 

on behalf of

Professor Robert Daniel Blank 

Academic Editor

PLOS ONE